# IMO: Greedy Layer-Wise Sparse Representation Learning for Out-of-Distribution Text Classification with Pre-trained Models

## Abstract

Machine learning models have made incredible progress, but they still struggle when applied to examples from unseen domains. This study focuses on a specific problem of domain generalization, where a model is trained on one source domain and tested on multiple target domains that are unseen during training. We propose IMO: **I**nvariant features **M**asks for **O**-of-Distribution text classification, to achieve OOD generalization by learning invariant features. During training, IMO would learn sparse mask layers to remove irrelevant features for prediction, where the remaining features keep invariant. Additionally, IMO has an attention module at the token level to focus on tokens that are useful for prediction. Our comprehensive experiments show that IMO substantially outperforms strong baselines in terms of various evaluation metrics and settings.

## 1 Introduction

When deploying natural language processing (NLP) models trained on labeled data in the wild, it is well known that they suffer from poor predictive performance on the samples drawn from the distributions different than their training (Wang et al., 2021b). This is due to the fact that the majority of NLP models assume that training and test data are identically and independently distributed (*i.i.d.*) (Schölkopf et al., 2021). Although various domain adaptation techniques have been proposed to bridge the gaps between training and testing distributions (Liu et al., 2022; Saunders, 2022), they all assume that labeled or unlabeled data from target domains is available during training and the domain information is known during testing. However, for many real-world applications, especially for early-stage businesses, their users may apply their models to arbitrary data such that the test data may well be Out-of-Distribution (OOD) and the domain information is not available for domain adaptation. In addition, their training datasets are often expensive to acquire so that they are available only in one domain. Therefore, this work focuses on single-source *domain generalization* (DG) for text classification, which aims to enable classifiers trained in *one* source domain to *robustly* work on the same classification tasks in any unseen OOD data without any model tuning.

Pre-trained large language models (LLMs) have drawn a lot of attentions recently due to their strong predictive performance on a variety of tasks. Although generative models or classifiers built on top of pre-trained LLMs outperform prior models in multiple domains, their performance is still not *robust* on the tasks, e.g. classification, when the testing distribution differs substantially from the training distribution (Bang et al., 2023). Recent works (Wang et al., 2021a; Feng et al., 2023; Veitch et al., 2021) show that one of the key reasons behind this is *spurious correlations*, which refer to the correlations between features and model outputs that are not based on causal relationships.

To take a step towards the goal "train it once, apply it anywhere", we propose a novel greedy layer-wise **I**nvariant **M**asking technique for **O**OD text classification, coined IMO, which selects domain-invariant features and key token representations from appropriate layers of a pre-trained deep transformer encoder to mitigate spurious correlations. The resulting hidden representations are sparse from the top layer to a specific layer of the pretrained model. We demonstrate the effectiveness of this technique through theoretical justifications and extensive experiments. Similar to (Zhang et al., 2021) on computer vision tasks, we shed light on how to apply sparsity as an effective inductive bias to deep pre-trained models for OOD text classification. Our contributions are summarized as follows:

- We propose IMO, a novel top-down greedy layer-wise sparse representation learning technique for pre-trained deep text encoders for robust OOD text classification by sharply reducing task-specific spurious correlations. In comparison with bottom-up layer-wise and simultaneous search across all layers, we discover that the top-down greedy search is decisive for performance improvement.

- We develop a theoretical framework that elucidates the relationship between domain-invariant features and causal features. Additionally, we provide an explanation of how our method learns invariant features.

- Our comprehensive experimental results demonstrate that

  - application of IMO to BART (Lewis et al., 2020), significantly outperforms competitive baselines, including recent LLMs, e.g. CHATGPT, on the classification of topics and sentiment polarity in the majority of the target domains, where CHATGPT has 10 times more parameters than BART;

  - application of IMO to CHATYUAN (Clue-AI, 2023) for Chinese also achieves superior performance over strong competitors, e.g. CHATGPT, on social factor classification;

  - spurious correlations in pre-trained models are harmful for OOD text classification;

  - IMO achieves similar OOD performance w.r.t. varying size of training data. The differences of accuracy between using 1k and 3.5 million training instances are less than 6%. In contrast, the corresponding accuracy differences of its backbone model without IMO is more than 16%.

## 2 RELATED WORK

**Domain Generalization.** Numerous DG methods have been proposed in the past decade, and most of them are designed for multi-source DG Chattopadhyay et al. (2020); Zhao et al. (2020); Ding et al. (2022); Zhang et al. (2022); Lv et al. (2022). Existing DG methods can be roughly classified into two categories: invariant representation learning and data augmentation. The key idea of the former category is to reduce the discrepancy between representations of source domains Muandet et al. (2013); Li et al. (2018a;b); Shao et al. (2019); Arjovsky et al. (2020). The key idea of data augmentation is to generate out-of-distribution samples, which are used to train the neural network with original source samples to improve the generalization ability Xie et al. (2020); Wei & Zou (2019); Volpi & Murino (2019). This paper focuses on single-source DG, where the model is trained on a single source domain, then evaluated on multiple unseen domains. Data augmentation is an effective strategy for single-source DG. Wang et al. (2021c) proposes a style-complement module to synthesize images with unseen styles, which are out of original distributions. Qiao et al. (2020) proposes adversarial domain augmentation to encourage semantic consistency between the augmented and source images in the latent space. Ouyang et al. (2023) uses a causality-inspired data augmentation approach to encourage network learning domain-invariant features. In terms of text classification, Ben-David et al. (2022); Jia & Zhang (2022) apply prompt-based learning methods to generate a prompt for each sample, then feed the prompt to a language model, finally predict labels based on vocabulary distribution.

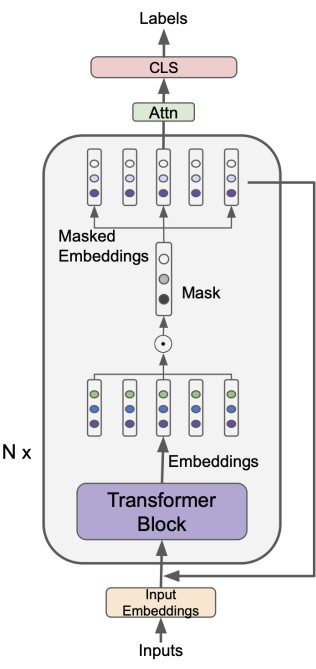

Figure 1: The overall architecture of our method IMO.

**Causal Representation Learning.** Causal representation learning addresses OOD generalization by exploring causal features that lead to labels. The assumption behind it is causal features are stable across different environments or data selections. Since causal representation learning is very ambitious and even infeasible in real application, a more practical method is invariant representation learning. Peters et al. (2016) investigated that invariant features, to some extent, infer the causal structure. Arjovsky et al. (2020) also assumes that prediction conditioned on invariant features is stable under different environments. Following such assumption, a strand of methods tries to

learn invariant features by mitigating spurious correlated features, which vary across environments Muandet et al. (2013); Chattopadhyay et al. (2020); Asgari et al. (2022); Izmailov et al. (2022); Hu et al. (2022b). This paper also follows this thread of methods, where we treat features that don't affect prediction as spurious correlated features. We use sparsification techniques Liu et al. (2020); Kusupati et al. (2020) to mask out spurious correlations.

## 3 METHODOLOGY

LLMs are pre-trained on large-scale corpora so that they can capture rich correlations between tokens across various domains. To enable trained models incorporating LLMs to work across domains, our key idea originates from the *Invariance Assumption* that the conditional distributions of labels conditioned on invariant features do not change across domains (Peters et al., 2016). Zhang et al. (2021) show that the assumption can hold, and there is a subnetwork inside a full network that can achieve better OOD performance than the full network. For a specific classification task, such as sentiment polarity analysis, the assumption indicates that there are certain sparse representations that are potential *causes* of labels (Wang & Jordan, 2022) across domains. Our method IMO realizes this idea by constructing sparse domain-invariant representations from the hidden representations of the selected layers of pre-trained transformer-based encoders.

Let $\mathcal{X}$ be the input space and $\mathcal{Y}$ be the label space, a *domain* is characterized by a joint distribution $P_{XY}$ on $\mathcal{X} \times \mathcal{Y}$. In the context of a single source DG, we have access to the data of one source domain $\mathcal{S} = \{(x^s, y^s)\}$ drawn from its joint distribution $P_{XY}^{\mathcal{S}}$. The goal is to learn a predictive model $f : \mathcal{X} \to \mathcal{Y}$ using only the data sampled from $P_{XY}^{\mathcal{S}}$ to minimize the prediction error on $K$ unseen target domains $\mathcal{T} = \{T_k = \{x^k\}\}_{k=1}^{K}$, each of which is associated with a joint distribution $P_{XY}^{(k)}$. Due to domain drifts, $P_{XY}^{\mathcal{S}} \neq P_{XY}^{(k)}, \forall k \in 1, ..., K$.

Following (Quinzan et al., 2023), we make the same assumptions that that (i) $Y = f(\text{Pa}(Y)) + \epsilon$, (ii) $\epsilon$ is exogenous noise, independent of any features, and (iii) $Y$ has no direct causal effect on any features, where $\text{Pa}(Y)$ denote parents of $Y$ in the underlying causal graph. Although $P_{XY}^{\mathcal{S}} \neq P_{XY}^{(k)}, \forall k \in 1, ..., K$, we show in Sec. 3.3 that under all above assumptions, there is a sparse representation $\mathbf{H}_i$ such that the function $Y = f(\mathbf{H}_i) + \epsilon$ exists in both source and target domains. The presence of invariant representations and influence of spurious correlations are empirically studied in Sec. 4.4.

As illustrated in Figure 1, our method constructs sparse domain-invariant representations at both feature and token levels in a top-down manner. At the feature level, given embeddings produced by the transformer block of the top layer, a parametric mask layer identifies invariant features from the embeddings. Then, the mask layer is frozen and the algorithm learns the mask layer for the lower layer. The process is repeated until a pre-specified layer is reached. At the token level, a soft attention mechanism incorporates the selected features from the top layer to identify key tokens and create aggregated sparse representations using only the invariant features for binary classification. For multi-class classification tasks, a sparse representation is created for each class so that each of them can focus on class-specific information. The model is regularized during training to increase the divergences of the representations between classes.

### 3.1 EXTRACTION OF INVARIANT FEATURES

Given a text input $X = [x_i]_{i=0}^{T}$, where $x_i$ is a token in $X$, a transformer-based pre-trained language model is employed to convert $x_i$ to a continuous token representation. We use hidden states produced by each transformer layer $l$ as token representations, denoted as $\boldsymbol{H}^l = [\boldsymbol{h}_i^l]_{i=0}^{T}$. $\boldsymbol{h}_i^l$ embeds both invariant features (useful for prediction in different domains) and spuriously correlated features (irrelevant for prediction) produced by layer $l$. Based on the Invariance Assumption, the invariant features $\boldsymbol{h}^*$ ensure $p^e(Y|\boldsymbol{h}^*)$ to be the same for all $e \in \mathcal{E}$, where $\mathcal{E}$ represents domains. In a transformer layer $l$, the spuriously correlated features are filtered out by performing element-wise multiplication between token representation $\boldsymbol{h}_i^l$ and a learnable mask $\boldsymbol{m}^l$.

A mask layer $\mathbf{m} = \mathbf{r} \odot \mathbf{q}$ contains zero and non-zero elements, where we define a trainable weight vector $\mathbf{r} \in \mathbb{R}^d$ and a trainable pruning threshold vector $\mathbf{s} \in \mathbb{R}^d$. A unit step function $g(t) = \begin{cases} 0 & \text{if } t < 0 \\ 1 & \text{if } t \geq 0 \end{cases}$ is applied to get a binary mask $\mathbf{q} = g(|\mathbf{r}| - \mathbf{s})$. By applying element-wise

multiplication $\mathbf{e}_i^l = \mathbf{h}_i^l \odot \mathbf{m}^l$, the zero elements of $\mathbf{m}$ remove corresponding features in token embeddings $\mathbf{h}^l$, while non-zero elements characterize the importance of corresponding features.

As the unit step function $g$ is not differentiable, we approximate its derivative by using the derivative estimator proposed in Xu & Cheung (2019) such that all parameters of a mask layer are trainable by using back-propagation and the family of stochastic gradient descent algorithms.

$$\frac{d}{dt}g(t) = \begin{cases} 2 - 4|t|, & -0.4 \leq t \leq 0.4 \\ 0.4, & 0.4 \leq |t| \leq 1 \\ 0, & \text{otherwise} \end{cases} \tag{1}$$

Following Xu & Cheung (2019), we add a sparse regularization term $L_{sparse}$ to the training loss to encourage the sparsity of mask layers:

$$\mathcal{L}_{sparse} = \sum_{i=1}^{N} \exp(-\boldsymbol{s}_i), \boldsymbol{s} \in \mathbb{R}^d \tag{2}$$

where $\exp(-\boldsymbol{s}_i)$ encourages high thresholds but prevents them from being extremely large. A higher threshold leads to removal of more features. During *inference*, we retain the mask layers to retain invariant features while discarding irrelevant ones.

## 3.2 IDENTIFICATION OF INVARIANT TOKENS

Given a long token sequence, not all information is useful for target tasks. For example, function words, such as 'the', or 'that', provide little information for predicting sentiment polarity. Thus, we employ a token-level attention mechanism to focus on important tokens. Instead of using all features of a token representation, we compute attention scores by using only the identified invariant features. The proposed attention mechanism differs slightly between binary and multi-class classification.

**Binary Classification.** For binary classification, we treat the mask vector $\mathbf{m}^L$ from the last layer $L$ as the query vector and compute the attention weight by performing the matrix product between $\mathbf{m}^L$ and each token embedding from the last layer $\mathbf{e}_i^L$: $a_i = \mathbf{m}^L \mathbf{e}_i^L$. Here, the mask vector and token embeddings are interpreted as matrices, with $\mathbf{m}^L \in \mathbb{R}^{1 \times d}$ and $\mathbf{e}_i^L \in \mathbb{R}^{d \times 1}$. For an input token sequence, we aggregate the masked token embeddings to obtain a sequence representation $\mathbf{v} = \sum_i^T a_i \mathbf{e}_i^L$, where $\mathbf{v} \in \mathbb{R}^{1 \times d}$. Finally, the sequence representation is fed into a fully-connected layer, followed by generating a distribution over the label space as follows: $\hat{\mathbf{y}} = \text{softmax}(\mathbf{v}\mathbf{P})$.

**Multi-class Classification.** For the multi-class classification task, we propose using multiple mask layers $\boldsymbol{m}_y^L$ in the last layer $L$ to capture corresponding features and tokens for labels $\boldsymbol{y}$. The number of mask layers equals the number of labels. Each label has its own attention weights $\boldsymbol{a}_y^L = \boldsymbol{m}_y^L \boldsymbol{e}$, and its own representation $\boldsymbol{v}_y^L = \sum_i^T a_{yi}^L \boldsymbol{e}_i$. Instead of using a fully-connected layer, we use a learnable weight vector per class to project $\boldsymbol{v}^L$ to a scalar: $c^L = \boldsymbol{v}^L \boldsymbol{p}^L$, where $\boldsymbol{v}^L \in \mathbb{R}^{1 \times d}$ and $\boldsymbol{p}^L \in \mathbb{R}^{d \times 1}$. The rationale behind this is that each class should have its own weight vector and hidden representations for encoding class-specific information. Then, we concatenate these scalars to a vector $\boldsymbol{c} = [c^L]$, and compute the predictive distribution by $\hat{\boldsymbol{y}} = \text{softmax}(\boldsymbol{c})$.

To encourage mask layers to extract label-specific features, we propose the following regularization term to penalize pairwise cosine similarities between the corresponding mask layers:

$$\mathcal{L}_{dist} = \frac{1}{N(N-1)} \sum_{i \neq j} \cos(\boldsymbol{m}^i, \boldsymbol{m}^j) \tag{3}$$

where $N$ is the number of label-specific mask layers.

**Training Procedure.** Rather than training all mask layers simultaneously, we adopt a layer-wise training procedure to train them sequentially from the top layer to the bottom layer. As illustrated in Figure 1, for each layer, a new mask layer, $\mathbf{m}^{L-i}$, is introduced on the top of the $(L-i)$-th transformer layer, with $i \in \{0, 1, 2, ...L-1\}$. Crucially, during this phase, the previously trained mask layers remain frozen to preserve their learned parameters. Upon each layer's training completion, the model is stored as $\theta_{L:L-i}$. This iterative procedure continues until the training of the most bottom mask layer, $\mathbf{m}^1$, is completed. Consequently, a suite of models, ranging from $\theta_L$ to $\theta_{L:1}$, is collected. We

empirically determine the model's efficacy by evaluating its performance on the validation set from the source domain. The best-performing model is chosen as the model to test on the target domains.

**Objective Function.** During training, the overall objective for binary classification is to (1) have good predictive performance on classification tasks and (2) maximize sparsity in mask layers to only keep invariant features. When training mask at layer $l$, the loss function is:

$$\mathcal{L} = \mathcal{L}_{ce} + \alpha \mathcal{L}^l_{sparsity} \tag{4}$$

where $\mathcal{L}_{ce}$ denotes standard cross entropy loss and $f$ denotes the predictive model. $\alpha$ is a hyperparameter that controls the balance between predictive performance and sparsity in mask layers. $\mathcal{L}^l_{sparsity}$ is the sparse regularization term for mask at layer $l$.

For multi-class classification, we add a distance regularization term:

$$\mathcal{L} = \mathcal{L}_{ce} + \alpha \mathcal{L}^l_{sparsity} + \beta \mathcal{L}_{dist} \tag{5}$$

The hyperparameter $\beta$ serves to calibrate the equilibrium between features specific to individual labels and those shared across multiple labels.

### 3.3 THEORETICAL ANALYSIS

Based on our assumptions, $Y = f(\mathbf{H}_i) + \epsilon$ exists, when $\mathbf{H}_i$ are the parent nodes of $Y$ in the underlying causal graph. Because $\mathbf{H}_i$ are a subset among all possible hidden representations correlated with $Y$, there should be a subset of hidden representations serving as parents of $Y$, otherwise the invariance assumption does not hold. Due to the widely used faithfulness assumption stating that statistical independences imply the corresponding causal structures (Neal, 2020), we aim to find out $\mathbf{H}_i \not\perp Y | \mathbf{H}_j$, where $\mathbf{H}_j$ is any feature set non-overlapped with $\mathbf{H}_i$.

We start our theoretical analysis by introducing a sparsity regularization term $\Omega(Y, H_i, ..., H_j)$, which counts the number of edges between $Y$ and the random variables of features in a underlying causal graph, where $Y$ is the variable for labels and $H_k$ denotes the random variable of the feature $h_k$. Then we introduce a loss function $\mathcal{L}_\Omega(Y, H_i, ..., H_j) = \mathcal{L}_{ce} + \alpha \Omega(Y, H_i, ..., H_j)$, analog to Eq. (4).

Considering the simplest case that there is only a causal feature $h_i$ and a non-causal feature $h_j$, the corresponding random variables are denoted by $H_i$ and $H_j$. From any causal graphs in Fig. **??**, we conclude that $p(Y|H_i, H_j) = p(Y|H_i)$ so that the cross entropy term in $\mathcal{L}_\Omega$ remains the same when using the term $p(Y|H_i)$, but the loss decreases after removing the non-causal feature from the loss due to the regularization term $\Omega(Y, H_i, H_j)$.

The two feature case can be easily extended to the case having more than two features. It is trivial that excluding a non-causal feature from the loss $\mathcal{L}_\Omega$ leads to the decrease of $\mathcal{L}_\Omega$ due to the Markov property of causal graphs (Peters et al., 2017).

**Corollary 1.** *If there is no edge between $Y$ and $H_k$ in a causal graph, then $\mathcal{L}_\Omega(Y, H_i, ..., H_j) < \mathcal{L}_\Omega(Y, H_i, ..., H_j, H_k)$.*

During training, we start with a loss $\mathcal{L}_\Omega(Y, H_1, ..., H_N)$ with a complete set of features. If a non-causal feature $H_k$ is removed, $\mathcal{L}_\Omega(Y, H_i, ..., H_j)$ decreases according to Corollary 1. In contrast, if a causal feature $H_k$ is removed, the cross entropy term increases because the mutual information $I(Y; H_k | H_i, ..., H_j) > 0$. Namely, $H_k$ adds additional information for predicting $Y$. However, in that case, $\mathcal{L}_\Omega(Y, H_i, ..., H_j)$ may still decrease if the increase of $\mathcal{L}_{ce}$ is smaller than the decrease of the regularization term $\alpha \mathcal{L}_\Omega(Y, H_i, ..., H_j)$. The exceptional case can be mitigated if $\alpha$ is sufficiently small. As a result, the loss $\mathcal{L}_\Omega$ provides an effective way to guide the search for the features serving as the causes of the labels, although we cannot recover the underlying true causal graphs. Herein, the loss (4) is a surrogate of $\mathcal{L}_\Omega(Y, H_i, ..., H_j)$ by using a deep neural network.

## 4 EXPERIMENTS

### 4.1 TASKS AND DATASETS

We evaluate our method on binary and multi-class classification. We evaluate the performance of the models using accuracy as the metric for binary classification tasks and macro-F1 as the metric

| Models | IMDB→ | | | Amazon→ | | | Yelp→ | | | TweetEval→ | | | Avg. |
|---|---|---|---|---|---|---|---|---|---|---|---|---|---|
| | Amazon | Yelp | TweetEval | IMDB | Yelp | TweetEval | IMDB | Amazon | TweetEval | IMDB | Yelp | Amazon | |
| BERT | 89.77* | 87.12* | 78.52* | 88.09* | 92.18* | 83.75* | 86.98* | 92.10* | 87.55* | 82.59* | 84.87* | 86.80* | 86.69* |
| BART | 89.91* | 88.01* | 68.47* | 87.93* | 91.01* | 82.98* | 86.44* | 91.97* | 88.21* | 78.21* | 89.51* | 87.01* | 85.80* |
| BERT-EDA | 87.73* | 87.47* | 72.10* | 88.89* | 92.43* | 86.40* | 88.11* | 92.98* | 87.92* | 81.64* | 85.82* | 87.77* | 86.61* |
| BERT-UDA | 87.76* | 87.02* | 70.23* | 89.87* | 93.78* | 86.37* | 86.89* | 92.81* | 84.91* | 82.83* | 85.95* | 87.29* | 86.31* |
| BERT-PGB | 88.40* | 83.61* | 70.51* | 89.70* | 93.66* | 86.19* | 86.09* | 92.72* | 87.95* | 81.88* | 85.13* | 87.54* | 86.11* |
| PADA | 85.73* | 89.84* | 88.40 | 84.47* | 93.96 | 85.92* | 87.71* | 91.42* | 90.33 | 80.30* | 84.69* | 90.61 | 87.78* |
| PDA | 89.35* | 90.59* | 87.71* | 88.16* | 94.20 | 85.61* | 88.17* | 93.59 | 89.88* | 82.05 | 86.37 | 86.41* | 88.51* |
| CHATGPT | 91.08 | 92.06 | 81.01 | 90.50 | 92.06 | 81.01 | **90.50** | 91.08 | 81.01 | **90.50** | 92.06 | 91.08 | 88.66 |
| ALPACA-7B | 90.14 | 92.30 | 88.66 | 83.01 | 92.30 | 88.66 | 83.01 | 90.14 | 88.66 | 83.01 | 92.30 | 90.14 | 88.52 |
| ALPACA-7B-LoRA | 89.80 | 82.80 | 87.77 | 81.00 | 82.80 | 87.77 | 81.00 | 89.80 | 87.77 | 81.00 | 82.80 | 89.80 | 85.34 |
| IMO-BART | **93.97** | **94.63** | **89.58** | **90.86** | **95.14** | **91.08** | 90.08 | **94.87** | **91.62** | 85.39 | **92.84** | 91.66 | **91.81** |
| IMO-BART B2T | 75.86* | 75.37* | 71.90* | 73.27* | 73.74* | 72.58* | 72.90* | 73.47* | 72.06* | 69.74* | 73.29* | 75.81* | 73.33* |
| IMO-BART w/o sq | 74.88* | 76.41* | 67.97* | 70.47* | 72.33* | 71.98* | 71.59* | 72.30* | 71.73* | 71.25* | 71.62* | 70.63* | 71.93* |
| IMO-BART last | 91.71* | 92.82* | 89.01 | 89.41 | 93.01* | 89.85* | 89.67 | 93.51 | 90.10* | 84.69* | 91.22* | 90.95* | 90.49* |

Table 1: Single-source domain generalization evaluation on sentiment analysis datasets. "B2T" signifies bottom-up layer-wise search. **w/o sq** indicates simultaneous search. "last" refers to only applying the mask on the last layer. PADA and PDA results are reproduced in our experiments since the original study evaluated performance on different datasets. Accuracy is the metric for evaluation. Asterisk * represents a significant difference compared to IMG-BART using a t-test with a $p \leq 0.05$.

for multi-class classification tasks. To assess the statistical significance of our results, we trained the models using five distinct random seeds.

The binary sentiment analysis datasets include Amazon Review Polarity Zhang et al. (2015b), Yelp Review Polarity Zhang et al. (2015b), IMDB Maas et al. (2011), TweetEval Sentiment Barbieri et al. (2020) and Yahoo! Answers Sentiment Li et al. (2019). TweetEval dataset has three categories: positive, negative, and neutral. We remove all neutral instances to align with other datasets. AG News Gulli (2005); Del Corso et al. (2005); Zhang et al. (2015a) is a collection of news articles for topic classification tasks. AG News contains news titles, and news descriptions, which belong to four topic classes. We train models on title phrases and test models on descriptions, and vice versa. SocialDial Zhan et al. (2023) is a Chinese socially-aware dialogue corpus consisting of two parts: synthetic conversation generated by CHATGPT and human-written conversations. Both are annotated with social factors such as location, social distance, and social relation. We train classification models on synthetic conversations and test models on human-written conversations. The statistics of the datasets can be found in Appendix A.1.

## 4.2 BASELINE MODELS

**Empirical Risk Minimization models.** As Gulrajani & Lopez-Paz (2021) showed that simple empirical risk minimization (ERM) outperforms many state-of-the-art domain generalization algorithms, we finetune **BERT** Devlin et al. (2019) and encoder of **BART** Lewis et al. (2020) using cross-entropy loss as two baselines. For Chinese text classification, we use **BERT-zh** Devlin et al. (2019), **BART-zh** Shao et al. (2021) and **CHATYUAN** Clue-AI (2023).

**Domain Generalization Models. PADA** Ben-David et al. (2022) is an example-based autoregressive prompt learning algorithm for domain generalization based on the T5 language model Raffel et al. (2020). Given an input, PADA first generates a prompt

| AG News | | | |
|---|---|---|---|
| Models | Title → Desc | Desc → Title | Avg-F1 |
| BERT | 81.11* | 67.95* | 74.68* |
| BART | 80.12* | 71.22* | 75.96* |
| BERT-EDA | 80.52* | 72.10* | 76.58* |
| BERT-UDA | 80.41* | 71.81* | 75.82* |
| BERT-PGB | 78.53* | 73.51* | 76,02* |
| PADA | 82.39* | 75.52* | 78.96* |
| PDA | 83.61* | 75.96* | 79.79* |
| CHATGPT | 85.13 | 79.28 | 82.21 |
| ALPACA-7B | 70.61 | 70.44 | 71.49* |
| ALPACA-7B-LoRA | 56.17 | 49.44 | 52.81* |
| IMO-BART | **89.40*** | **81.97*** | **85.68*** |
| IMO-BART B2T | 70.31* | 64.59* | 67.45* |
| IMO-BART w/o sq | 62.59* | 57.27* | 59.93* |
| IMO-BART last | 88.22 | 80.05* | 84.13* |

Table 2: Evaluation results on multi-class classification datasets. 'Desc' represents description. The metric is macro F1.

and then predicts the label of input concatenated with this prompt. **PDA** Jia & Zhang (2022) is a prompt-based learning algorithm for domain generalization, which applies vocabulary distribution alignment and feature distribution alignment to reduce the gap between different domains.

**Large Language Models.** As CHATGPT shows promising zero-shot ability on various NLP tasks OpenAI (2023), we treat **CHATGPT** (gpt-3.5-turbo model) as one baseline. It is worth noting that, as datasets for experiments are publicly available, we are uncertain if CHATGPT used these datasets during training. **ALPACA-7B** Taori et al. (2023) is another baseline, which is fine-tuned from the 7B LLaMA model Touvron et al. (2023) on 52K instruction-following data generated by self-instruct Wang et al. (2022). **ALPACA-7B-LoRA** is fine-tuned ALPACA-7B model using low-rank adaptation Wang (2023); Hu et al. (2022a). **CHATGLM-6B** THUDM (2023) is an open large language model based on General Language Model Du et al. (2022) framework, optimized for Chinese question-answering and dialogue. All LLMs use a few-shot in-context learning setting. The specific query templates adopted for LLMs can be found in Appendix A.2.

**Data Augmentation.** Wiles et al. (2022); Gokhale et al. (2022) find data augmentation benefit domain generalization tasks. Thus, we use three text data augmentation techniques as baselines. **EDA** Wei & Zou (2019) is a widely used text data augmentation technique, which consists of four simple but powerful operations: synonym replacement, random insertion, random swap, and random deletion. **UDA** Xie et al. (2020) use back-translation to generate diverse paraphrases while preserving the semantics of the original sentences. Shiri et al. (2023), henceforth referred to as **PGB** for brevity, generates syntactically and lexically diversified paraphrases using a fine-tuned BART.

**Ablation Study Models.** Besides baselines, we also apply ablation studies. Our method can apply to backbone models such as BART, T5, and BERT. To distinguish these specific implementations, these implementations are referred to as **IMO-BART**, **IMO-T5**, and **IMO-BERT**. To clarify the contribution of each component in our model, we conduct additional experiments where we exclude mask layers and attention mechanisms. These modified variants are denoted as **w/o $m$**, **w/o $a$**, **w/o $am$** indicating the removal of mask layers, attention mechanisms, and both components respectively. Additionally, we employ various sparsification techniques to implement sparse layers, namely, **STR** Kusupati et al. (2020), **STE** Bengio et al. (2013); Liu et al. (2020), and **Scalar**, where a learnable single scalar is used instead of the threshold vector $s$. Training details can be found in Appendix A.2.

| | SocialDial | | | |
|---|---|---|---|---|
| **Models** | **Loc (Synthetic) $\rightarrow$ Loc (Human)** | **SD (Synthetic) $\rightarrow$ SD(Human)** | **SR (Synthetic) $\rightarrow$ SR(Human)** | **Avg- F1** |
| BERT-zh | 18.11* | 35.05* | 32.39* | 28.51* |
| CHATYUAN | 18.23* | 34.94* | 33.92* | 29.03* |
| BERT-EDA | 13.98* | 35.71* | 26.38* | 25.36* |
| BERT-UDA | 15.20* | 33.59* | 27.03* | 25.27* |
| CHATGPT | 21.44 | 38.46 | 35.12 | 31.67 |
| CHATGLM-6B | 20.57 | 20.53 | 11.55 | 17.55 |
| IMO-CHATYUAN | **23.22** | **46.04** | **42.71** | **37.32** |
| IMO-CHATYUAN B2T | 14.31* | 30.29* | 32.45* | 25.68* |
| IMO-CHATYUAN w/o **sq** | 13.37* | 29.81* | 29.05* | 24.07* |
| IMO-CHATYUAN last | 21.47* | 44.73 | 39.89* | 35.36* |

Table 3: Evaluation results on SocialDial dataset. Loc represents Location; SD represents Social Distance; SR represents Social Relation. The metric is macro F1.

| | IMDB$\rightarrow$ | | | Amazon$\rightarrow$ | | | Yelp$\rightarrow$ | | | TweetEval$\rightarrow$ | | | |
|---|---|---|---|---|---|---|---|---|---|---|---|---|---|
| **Models** | **Amazon** | **Yelp** | **TweetEval** | **IMDB** | **Yelp** | **TweetEval** | **IMDB** | **Amazon** | **TweetEval** | **IMDB** | **Yelp** | **Amazon** | **Avg.** |
| IMO-BART | **93.97** | **94.63** | **89.58** | **90.86** | **95.14** | **91.08** | **90.08** | **94.87** | **91.62** | 85.39 | **92.84** | **91.66** | **91.81** |
| IMO-T5 | 93.45 | 93.88 | 84.92* | 89.23* | 93.38* | 89.73* | 88.27* | 93.02* | 91.01 | 81.39* | 91.93 | 89.97* | 90.01* |
| IMO-BERT | 86.10* | 81.73* | 77.41* | 81.35* | 84.69* | 82.08* | 80.96* | 87.79* | 86.51* | 78.13* | 80.20* | 84.13* | 82.59* |
| IMO-BART w/o $a$&$m$ | 89.94* | 89.13* | 69.59* | 88.19* | 92.20* | 82.69* | 86.85* | 90.64* | 85.83* | 78.98* | 89.25* | 87.58* | 85.91* |
| IMO-BART w/o $m$ | 92.15* | 92.49* | 85.61* | 89.48* | 92.97* | 88.53* | 88.28 | 92.75 | 87.44 | 80.10 | 89.57 | 88.09* | 88.95* |
| IMO-BART w/o $a$ | 91.35* | 91.04* | 84.18* | 88.51* | 92.49* | 84.97* | 87.10* | 91.87* | 88.01* | 83.31* | 90.61* | 88.87* | 88.52* |
| IMO-BART STE | 91.11* | 91.71* | 88.05* | 88.29* | 91.69* | 87.09* | 88.91* | 91.39* | 89.12* | 82.48* | 89.37* | 88.50* | 88.97* |
| IMO-BART STR | 89.79* | 88.97* | 72.98* | 86.26* | 87.48* | 79.48* | 86.40* | 88.31* | 77.49* | 81.43* | 85.13* | 82.49* | 83.85* |
| IMO-BART Scalar | 87.31* | 89.92* | 87.34* | 87.73* | 86.03* | 83.41* | 87.11* | 86.43* | 85.94* | 81.44* | 84.75* | 85.41* | 86.06* |

Table 4: Ablation study on sentiment analysis datasets.

## 4.3 DOMAIN GENERALIZATION RESULTS

Tables 1, 2, 3 present our results. We report the accuracy score for binary classification (*i.e.,* sentiment analysis) and the macro-F1 score for multi-class classification.

**Results on Binary Classification** are presented in Table 1, where our method using BART as backbone (*i.e.,* IMO-BART) outperforms all baselines in 7 of 12 settings. In terms of average accuracy, IMO-BART reaches the highest result, exhibiting average performance gains of 2.63% than the best baseline (*i.e.,* CHATGPT). Interestingly, CHATGPT stands out as the leading model in 3 out of 12 settings. Notably, ALPACA-7B demonstrates a performance level matching with CHATGPT. ALPACA-7B-LoRA also shows relatively good performance across all settings. In

| AG News | | | |
|---|---|---|---|
| **Models** | **Title → Desc** | **Desc → Title** | **Avg-F1** |
| IMO-BART | **89.40** | **81.97** | **85.68** |
| IMO-T5 | 86.91* | 79.75* | 83.33* |
| IMO-BERT | 84.79* | 75.38* | 80.09* |
| IMO-BART w/o $a\&m$ | 80.91* | 73.89* | 77.40* |
| IMO-BART w/o $m$ | 83.29* | 77.08* | 80.19* |
| IMO-BART w/o $a$ | 82.72* | 77.27* | 79.99* |
| IMO-BART Binary | 87.79* | 79.82* | 83.81* |

Table 5: Ablation study on AG News dataset. 'Binary' refers to the application of the proposed binary classification method on multi-label classification tasks. The evaluation metric is macro F1.

comparison, it remains uncertain whether CHATGPT has been trained on the datasets used in this paper. To the best of our knowledge, it is unlikely that ALPACA-7B and ALPACA-7B-LoRA have been trained on these datasets. These observations indicate the strong generalization ability of large language models. Moreover, it is noteworthy that data augmentation methods (*i.e.,* BERT-EDA, BERT-UDA, BERT-PGB) show slightly inferior performance in comparison to the simple fine-tuning of BERT in terms of average accuracy. This suggests that simply back-translating or paraphrasing instances within source domains does not enhance performance on target domains. A plausible explanation could be that augmented data inherently belongs to source domains. Thus, while training on such data might improve performance in source domains, it is unlikely to increase performance in unseen target domains.

In addition, we find out that our method has better performance than bottom-up layer-wise search (B2T), simultaneous search (w/o **sq**), and only applying a mask on the last layer (last). This indicates that the top-down greedy search is crucial for performance improvement.

**Results on Multi-class Classification** are presented in Table 2 and 3. Our method outperforms all baselines in terms of average macro-F1 by 3.22% and 5.16% on AG News and SocialDial respectively. Among baselines, CHATGPT exhibits the strongest performance on both datasets and surpasses ALPACA-7B, ALPACA-7B-LoRA, and CHATGLM by a large margin. This superior performance shows that current open-source large language models still have a substantial performance gap with CHATGPT when handling difficult tasks.

| SocialDial | | | | |
|---|---|---|---|---|
| **Models** | **Loc (Synthetic) → Loc (Human)** | **SD (Synthetic) → SD(Human)** | **SR (Synthetic) → SR(Human)** | **Avg- F1** |
| IMO-CHATYUAN | **23.22** | **46.04** | **42.71** | **37.32** |
| IMO-BART-zh | 19.94* | 41.39* | 39.27* | 33.53* |
| IMO-BERT-zh | 14.68* | 36.75* | 27.41* | 26.28* |
| IMO-CHATYUAN w/o $a\&m$ | 19.12* | 37.75* | 34.07* | 30.31* |
| IMO-CHATYUAN w/o $m$ | 22.47* | 41.86* | 38.95* | 34.43* |
| IMO-CHATYUAN w/o $a$ | 21.05* | 39.88* | 37.28* | 32.73* |
| IMO-CHATYUAN w/o Binary | 20.17* | 39.26* | 39.41* | 32.95* |

Table 6: Ablation study on SocialDial datasets.

## 4.4 ANALYTICAL EXPERIMENTS

**Presence of Invariant Representations.** We visualize the masks of the top layer in multiple domains in Fig. 3. A close investigation shows that there are indeed several features shared across domains. We further compute Cosine similarities between the masks of the top layer trained on different source domains. As shown in Table 10, the similarities between masks range from 0.68 and 0.85. The token-level sparsity is illustrated by the attention weights visualized in Fig.2 to show the presence of shared key words across domains.

**Impact of Spurious Correlations in Classification.** To study whether our proposed masking mechanism indeed identifies robust features, we conduct experiments where we replace the learned

| Models | Yelp→ | | | | Amazon→ | | | |
|---|---|---|---|---|---|---|---|---|
| | Yelp (Source) | IMDB (Target) | Amazon (Target) | TweetEval (Target) | Amazon (Source) | IMDB (Target) | Yelp (Target) | TweetEval (Target) |
| IMO-BART | 95.94 | -5.86 | -1.07 | -4.32 | 95.34 | -4.48 | -0.20 | -4.26 |
| IMO-BART- SC | 89.01* | -7.81* | -3.88* | -11.20* | 90.12* | -7.64* | -3.47* | -12.59* |

Table 7: Comparison between the proposed model and model using spurious correlation features (SC). In target datasets, we report the reduced percentage of accuracy compared to the source domains. The asterisk * in the reverse mask model represents a significant difference compared to the performance of our proposed model.

binary masks $\mathbf{q}$ by $|1 - \mathbf{q}|$. We freeze all parameters except the classification head, then train a model using spurious features identified by $|1 - \mathbf{q}|$. The results in Table 7 show that models using spurious features have an approximate 6% accuracy reduction in source domains and perform also worse than using all features. In target domains, the corresponding performance decrease using spurious features is significantly higher than both our method and the models using all features.

**Ablation Study.** We compare variants of IMO and report the results in Table 4, 5, 6. Compared with variants that remove both the attention module and mask layers, IMO with the attention module or mask module has a performance improvement in terms of average accuracy or average F1, which suggests the two modules promote performance mutually. Moreover, we replace different backbone models. The experiment results suggest that encoder-decoder pre-trained language models have (*i.e.,* BART, T5) present better performance than encoder-only language models (*i.e.,* BERT).

Additionally, we compare IMO with various sparsity methods to implement mask layers. The experiment results are presented in Table 4. STE provides a distinct derivative estimation of the unit step function, exhibiting a marginally lesser impact on our approach. STR, another threshold-based method to implement sparsity in neural networks, results in a performance decline but can achieve higher sparsity in mask layers. Scalar, a variant of our method, substitutes threshold vectors with a single learnable scalar, severely reducing its capacity to achieve sparsity in mask layers.

**Training Data Size in Source Domains.** To explore the influence of source domain training data size on performance within target domains, we train models based on BART with and without our method on the Amazon review dataset with varying sizes of training data (*i.e.,* 1k, 10k, 100k, 1M, and 3.6M). The results in Table 8 show that our method depends significantly less on training data size, though more training data can improve the performance overall. Notably, 1k training data yields a remarkable decline for the models without using IMO, while the corresponding performance reduction is significantly less by using our method.

| Models | Amazon→ | | | |
|---|---|---|---|---|
| | Yelp | IMDB | TweetEval | Avg. |
| IMO-1k | 92.21 | 87.29 | 85.18 | 88.22 |
| IMO-10k | 94.82 | 89.11 | 88.43 | 90.78 |
| IMO-100k | 94.90 | 90.24 | 89.01 | 91.38 |
| IMO-1M | 94.95 | 90.29 | 89.20 | 91.48 |
| IMO-3.6M | 95.14 | 90.86 | 91.08 | 92.36 |
| IMO- w/o $am$ -1k | 70.62 | 68.61 | 66.07 | 68.43 |
| IMO- w/o $am$ -10k | 84.88 | 79.02 | 75.19 | 79.70 |
| IMO- w/o $am$ -100k | 87.05 | 84.95 | 80.48 | 84.16 |
| IMO- w/o $am$ -1M | 91.38 | 87.06 | 81.59 | 86.68 |
| IMO- w/o $am$ -3.6M | 92.20 | 88.19 | 82.69 | 87.69 |

Table 8: Domain generalization experiment with different training sizes in the source domain.

## 5 CONCLUSION

This paper presents the first work aiming to improve single-source domain generalization on pre-trained deep encoders for text classification tasks. Herein, we introduce a novel method called IMO, which is a greedy layer-wise representation learning method aiming to identify invariant features and token representations from multiple layers of the encoder. The key idea is to retain invariant features through trainable mask layers, and incorporate a token-level attention module to focus on the tokens that directly lead to the prediction of labels. In addition, we provide a theoretical explanation about why our method works from a causal perspective. Through extensive experiments, we demonstrate that IMO achieves superior OOD performance over competitive baselines on multiple datasets, if the pre-trained encoders provide strong domain-invariant features. The visualization of mask layers and attention weights empirically justifies the identified domain-invariant sparse representations.

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

# A APPENDIX

## A.1 EXPERIMENT DATASETS

The statistics of datasets are listed in Table 9.

| Binary Classification | | | | |
|---|---|---|---|---|
| **Dataset** | **Domain** | **#Train** | **#Dev** | **#Test** |
| Amazon | Review of products | 3.6M | 0 | 40k |
| IMDB | Review of movies | 25k | 0 | 25k |
| Yelp | Review of businesses | 560k | 0 | 38k |
| TweetEval | Tweet | 25k | 1k | 6k |
| Yahoo | Questions from Yahoo! Answers | 4k | 2k | 1k |

| Multi-class Classification | | | | |
|---|---|---|---|---|
| **Dataset** | **Domain** | **#Train** | **#Dev** | **#Test** |
| AG News | Title of news articles | 120k | 0 | 7k |
| AG News | Description of news articles | 120k | 0 | 7k |
| SocialDial | Synthetic conversations by CHATGPT | 68k | 7k | 7k |
| SocialDial | Human-written conversations | 0 | 0 | 5k |

Table 9: Statistics of datasets.

## A.2 TRAINING DETAILS

We use the encoder of BART Lewis et al. (2020) as the default pre-trained language model. All models are trained up to 100 epochs with a minibatch size 32 in the source domain. We use Adam (Kingma & Ba, 2015) optimizer with hyperparameters tuned on the validation sets. As a result, we run Adam with $\beta_1 = 0.9$ and $\beta_2 = 0.999$. The learning rate is $5 \times 10^{-5}$. We use a linear learning rate scheduler that dynamically decreases the learning rate after a warm-up period. All experiments are conducted on NVIDIA A40 GPU.

The process of model selection in domain generalization is inherently a learning problem. In this approach, we employ training-domain validation, which is one of the three selection methods introduced by Gulrajani & Lopez-Paz (2021). We divide each training domain into separate training and validation sets. Models are trained on the training set, and the model that achieves the highest accuracy on the validation set is chosen as the selected model.

| | Yelp | Amazon | IMDB | TweetEval |
|---|---|---|---|---|
| **Yelp** | 1.0 | 0.7930 | 0.7533 | 0.6838 |
| **Amazon** | - | 1.0 | 0.8458 | 0.7687 |
| **IMDB** | - | - | 1.0 | 0.8069 |
| **TweetEval** | - | - | - | 1.0 |

Table 10: Cosine similarities between mask vectors $m$ trained on different source domains.

| | Yelp | Amazon | IMDB | TweetEval |
|---|---|---|---|---|
| **Yelp** | 1.0 | 0.5869 | 0.5231 | 0.4504 |
| **Amazon** | - | 1.0 | 0.6513 | 0.5614 |
| **IMDB** | - | - | 1.0 | 0.6139 |
| **TweetEval** | - | - | - | 1.0 |

Table 11: Jaccard similarities between binary vectors $q$ trained on different source domains.

When using large language models to predict target classification labels, the query template for sentiment analysis is: "There are some examples about sentiment analysis: {examples}. Given text: {sentence}, what is the sentiment conveyed? Please select the answer from 'positive' or 'negative'.". The query template for AG News topic classification is "There are some examples for topic classification: {examples}. Given text: {sentence}, what is the topic of this text? Please select the answer from 'Business', 'Sci/Tech', 'World' or 'Sports'." The query templates for SocialDial are "There are some examples for classification: {examples}. Given conversation: {conversation}, what's the location/social distance/social relation of this conversation? Please select the answer from {labels}"[1] (Min et al., 2022; Wang et al., 2023; Yang et al., 2023).

## A.3 VISUAL EXPLANATION

To intuitively show how the attention module and mask module work in models, we visualize attention weights on tokens and mask vectors in Figure 2 and 3, respectively. We also demonstrate cosine similarities between mask vectors $m$ trained on different source domains and Jaccard similarities between binary vectors $q$ trained on different source domains on Table 10 and Table 11, respectively.

From Figure 2, we can find that our model primarily focuses its attention on sentiment-indicative tokens. Notably, positive reviews exhibit high attention weights for tokens like 'good,' 'great,' and 'nice,' indicating their significance. Conversely, negative reviews assign high attention weights to tokens such as 'horrible' and 'slow,' highlighting their importance in expressing negativity.

In Figure 3, we visualize mask vectors $m$ and binary vectors $q$ trained on different source domains across dimensions. It can be observed that certain dimensions are consistently assigned zero (or non-zero) values across different training domains, indicating our mask layers can capture some features that are irrelevant (or invariant) across domains. We quantify invariant features across domains by computing vector similarity. We calculate cosine similarities between different mask vectors $m$. The results are shown in Table 10. We can find that most mask vector pairs have over 0.75 similarity, except the Yelp-TweetEval pair, which is probably because of a larger divergence between Yelp and TweetEval domains. Table 11 shows Jaccard similarities between binary vectors $q$. Most binary vector pairs have similarities of over 0.5, except the Yelp-TweetEval pair, with a similarity of 0.45.

---

[1] Since SocialDial is a Chinese dataset, we provided a translated query from Chinese to English in order to enhance comprehension for large language models.

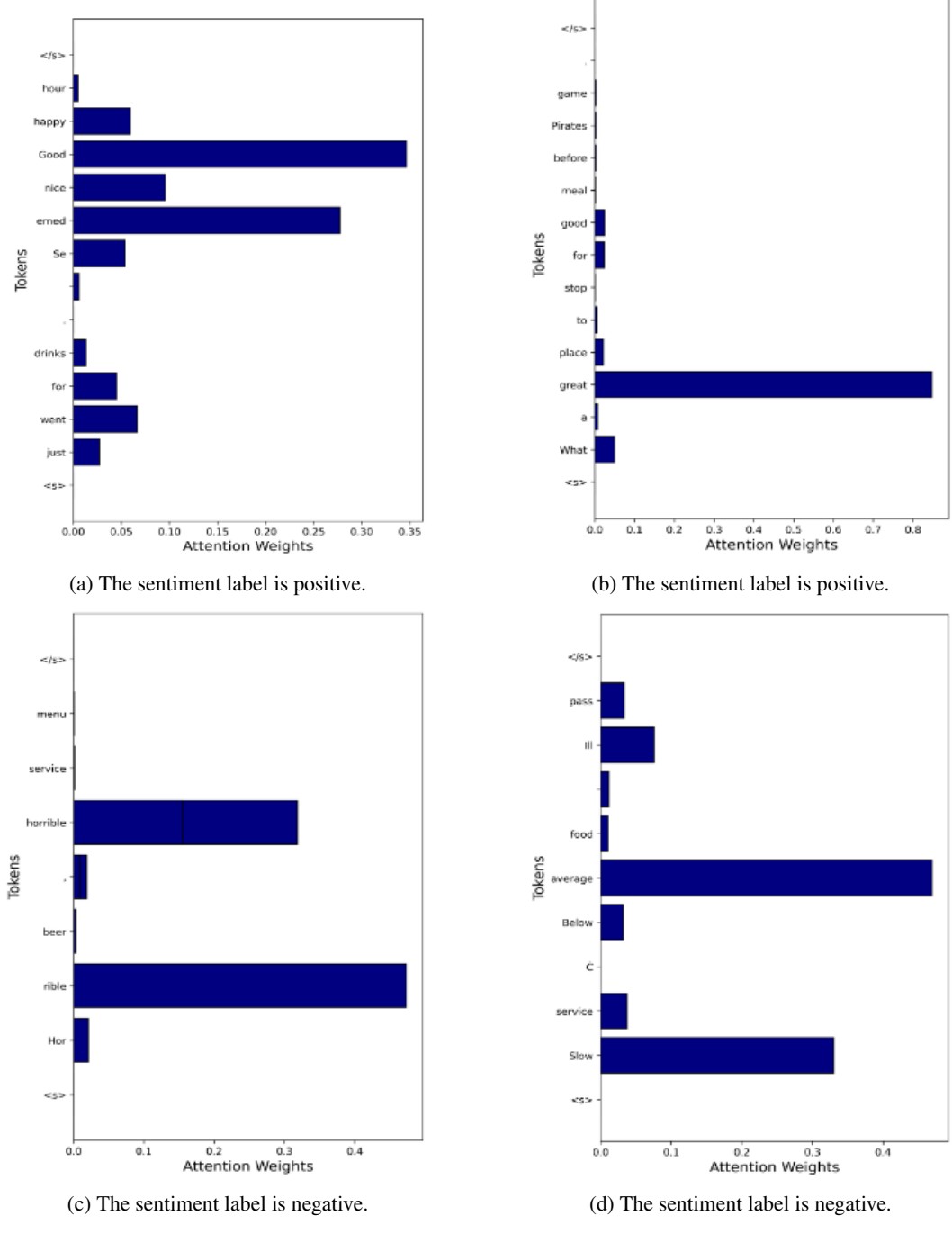

(a) The sentiment label is positive.

(b) The sentiment label is positive.

(c) The sentiment label is negative.

(d) The sentiment label is negative.

Figure 2: Visualization of attention weights on tokens in Yelp dataset reviews.

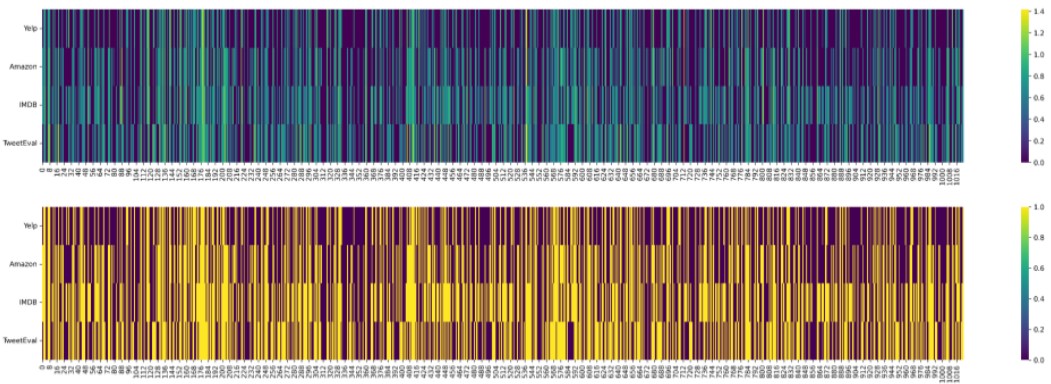

Figure 3: Visualization of mask layers in IMO-BART trained on sentiment analysis datasets. The top figure visualizes the mask vectors $m$, while the bottom one visualizes the binary vectors $q$. The x-axis signifies the dimensionality of mask layers, whereas the y-axis denotes values attributed to each dimension.

