# OpenReview forum: "IMO: Greedy Layer-Wise Sparse Representation Learning for Out-of-Distribution Text Classification with Pre-trained Models"
_ICLR.cc/2024/Conference — ICLR 2024 Conference Withdrawn Submission_

### Official Review · Reviewer_hBe7 · 2023-10-29

**Soundness:** 2 fair
**Presentation:** 2 fair
**Contribution:** 2 fair
**Rating:** 3
**Confidence:** 4

**Summary:**

The gist of the paper is that sparse constraints can help to alleviate OOD problem, and doing it in a top-down fashion is much better than the other way around.

**Strengths:**

1. Presents in a way that easy to follow the main issues.
2. The method is easy to understand.

**Weaknesses:**

1. Some lousy typos/mis-arrangement of the writing make the most important part not readable.
2. A mis-match between the motivating theory and the proposed method.
3. Experiments are not very convining.

**Questions:**

I have following major concerns:

1. I don't see why adding sparsity constraint doesn't necessarily provide you domain invariant feature. Sparsity only makes activation patterns sparse but somehow there is no guarantee those will be invariant over other domains. I think there is a huge gap between the motivating theory and the proposed method. To convince me, I believe you should empirically show that all the remaining non-sparse features are indeed invariant according to P_{X,Y} you defined.

2. Can you also perform the same method on BERT and ALPACA-7B? (i.e., IMO-BERT, IMO-ALPACA-7B)? Supposedly we should also see the improvement if the method is really working.

3. Can you add the variance bound as well for the experiments? For some dataset, it reads like the improvement is totally due to randomness of init rather than the method.

3. Your theory motivation apparently borrows from SCM based causality. But somehow I really didn't see any SCM figure in the paper. In particular, I only read:

"From any causal graphs in Fig. ??, we conclude that p(Y |Hi, Hj ) = p(Y |Hi) so that the cross entropy term in LΩ remains the same when ....",

It's quite unbelievable that the most important part of the theory is shown in Figure ??, so I could also just response in a way that my understanding is ??.

Without a clear make-up clarification, I will rate the draft as not ready just in terms of writing.

---

### Official Review · Reviewer_vuMm · 2023-11-10

**Soundness:** 2 fair
**Presentation:** 2 fair
**Contribution:** 2 fair
**Rating:** 3
**Confidence:** 4

**Summary:**

The paper studies the problem of domain generalization from one source domain to multiple unseen target domains. The paper proposes a new method, call IMO. Its main idea is to find sparse latent representations that are invariant across domains. Masks are used to disable/activate certain dimensions of the representation. Regularization terms are used to encourage sparsity. The proposed algorithm is evaluated empirically and shown effective.

**Strengths:**

1. The overall idea makes sense.
2. The experimental results look good.

**Weaknesses:**

1. The writing lacks clarity at places. Some details of the model are vague to this reviewer.
2. The work is largely heuristic and is based on intuitive argument. Theoretical justification is weak, serving mostly decorative purposes. Some of construction appear unjustified, for example, ${\cal L}_{dist}$.

To this reviewer, this paper belongs to the many works that have a reasonable idea. But the approach is not of great novelty and the work contains little insight at depth.

**Questions:**

All tasks considered are classification tasks, in which $Y$ is categorical. How is the assumption $Y=f(H_i)+\epsilon$ relevant?

---

### Official Review · Reviewer_aKhG · 2023-11-18

**Soundness:** 2 fair
**Presentation:** 2 fair
**Contribution:** 2 fair
**Rating:** 3
**Confidence:** 4

**Summary:**

This paper studies out-of-distribution text classification from a single source domain. The main idea is to learn sparse mask layers that filter out spurious features while retaining invariant features. The masks are learned through the joint optimization of three loss terms: a cross-entropy loss for classification, a sparsity loss that enforces sparse masks, and a cosine-similarity loss that encourages the mask layers to extract label-specific features. The network is trained in a sequential manner from the top layer to the bottom layer. Theoretical analysis shows that given a set of causal and non-causal features. The empirical evaluation is done on binary and multi-class classification. Compared with a variety of language models including LLMs (few-shot in-context), the proposed method demonstrates superior performance on several datasets.

**Strengths:**

- The studied problem is important to real-world applications of language models.
- The empirical results seem strong. A great variety of language models are compared with the proposed method on a number of datasets. The advantage of the proposed method is demonstrated clearly.
- The paper is overall well written and easy to understand.

**Weaknesses:**

- The connection between the theory and the actual problem being studied is somewhat weak. The theory assumes that all variables correlated with $Y$ are observed; however, this assumption may not hold in practice. For example, there may be an unobserved selection variable that introduces selection bias in the training domain by forming a collider with $Y$ and some non-causal variables $H_j$. In this case, $p(y|h_i, h_j) = p(y|h_i)$ does not hold. Removing $H_j$ may incur a higher cross-entropy loss than removing $H_i$.
- The basic idea of the proposed method seems to be largely based on [1]. The main difference between the two methods, in my point of view, is the target to which the mask is applied (token embedding v.s. model weight). It is unclear whether there is any fundamental difference between these two approaches.

[1] Zhang, Dinghuai, et al. "Can subnetwork structure be the key to out-of-distribution generalization?." International Conference on Machine Learning. PMLR, 2021.

**Questions:**

- Does the result of the current theory hold if there is some unobserved selection variable between $Y$ and some non-causal variables?
- Is there any fundamental difference between the proposed method and [1]? If so, what is it?
- What are the causal graphs referred to on page 5 with "Fig. ??"?